# Current Development in Interdigital Transducer (IDT) Surface Acoustic Wave Devices for Live Cell In Vitro Studies: A Review

**DOI:** 10.3390/mi13010030

**Published:** 2021-12-27

**Authors:** Mazlee Bin Mazalan, Anas Mohd Noor, Yufridin Wahab, Shuhaida Yahud, Wan Safwani Wan Kamarul Zaman

**Affiliations:** 1AMBIENCE, Faculty of Electronic Engineering Technology, Universiti Malaysia Perlis, Arau 02600, Perlis, Malaysia; anasnoor@unimap.edu.my (A.M.N.); yufridin@unimap.edu.my (Y.W.); shuhaidayahud@unimap.edu.my (S.Y.); 2Department of Biomedical Engineering, Faculty of Engineering, Universiti Malaya, Kuala Lumpur 50603, Selangor, Malaysia

**Keywords:** surface acoustic wave (SAW) technique, acoustofluidic, interdigital transducer (IDT), live cells

## Abstract

Acoustics have a wide range of uses, from noise-cancelling to ultrasonic imaging. There has been a surge in interest in developing acoustic-based approaches for biological and biomedical applications in the last decade. This review focused on the application of surface acoustic waves (SAW) based on interdigital transducers (IDT) for live-cell investigations, such as cell manipulation, cell separation, cell seeding, cell migration, cell characteristics, and cell behaviours. The approach is also known as acoustofluidic, because the SAW device is coupled with a microfluidic system that contains live cells. This article provides an overview of several forms of IDT of SAW devices on recently used cells. Conclusively, a brief viewpoint and overview of the future application of SAW techniques in live-cell investigations were presented.

## 1. Introduction

In live-cell studies, whether in vivo or in vitro, scientists and doctors are constantly inventing new approaches and methodologies to analyse specific cell characteristics and/or behaviours in multicellular or unicellular organisms, either in isolation or as part of a tissue. For example, the utilisation of biochemical, electric field, and mechanical stimuli has been extensively investigated in live-cell studies [1,2]. Mechanical stimulation has been widely accepted to influence cell structure, function, and development. Therefore, it has recently stimulated interest as a tool for manipulating, coordinating, organising, and characterising cell properties. For instance, mechanical stimuli such as cyclic stretching [3], fluid shear stress [4], electrical [5] and magnetic fields [6], substrate stimulation [7], and acoustic waves technique [8] have been used to address a variety of challenges in biomedical research, especially in the fields of clinical diagnostics and therapy.

The acoustic waves technique was firstly documented by White and Voltmer, with the introduction of the interdigitated transducer (IDT) in 1965 [9]. They showed that the IDT, which consists of spatially periodic thin-film metal electrodes over a quartz substrate, can generate travelling surface acoustic waves (SAW), as illustrated in Figure 1. By applying an alternating voltage (AC) to the SAW on a piezoelectric substrate with IDT, a periodic electric field is imposed on the piezoelectric substrate’s crystal. A standing surface acoustic wave is produced as a result of the generation of a periodic strain field. The IDT electrode must be properly constructed to provide an optimal SAW device because several parameters affect the SAW device performance, such as wavelength, aperture, number of fingers, and delay line. The device has attracted exceptional attention, particularly in the electronic industry, such as telecommunications and healthcare fields, for application in signal frequencies ranging from 10 MHz to 1 GHz [10].

In the 19th century, the application of the SAW device diverted to cellular biology research, increasing tremendously from 1990 as depicted in Figure 2, due to the several advantages of the SAW technique. First, SAW devices have a distinct advantage of being able to be used for both sensing and actuation. The SAW device can act as a sensor that offers a high level of sensitivity. Meanwhile, the application of acoustic waves within microfluidic, also known as acoustofluidic, can handle liquid quantities ranging from a few tens of microlitres to a few picolitres [11], and have been shown to have a variety of impacts on cell viability, proliferation, stress, and function [12]. Second, the SAW device is simple and easy to fabricate using a conventional photolithography process. Third, SAW is less likely to harm the cells and biomolecules. Fourth, SAW manipulates cells without contact with the biological targets, so that complications such as cell damage and biofouling are prevented [13].

This review discusses six major applications of the SAW techniques: cell manipulation, cell separation, cell seeding, cell migration and proliferation, cell mechanical properties, and cell behaviours, including the working mechanisms and key findings for each application. Unlike other review articles in the field of SAW that focused only on a specific application and used many types of devices [10,13,14,15,16,17], this article focuses on the implementation of SAW techniques by using IDT on various types of live cells. Furthermore, an overview of different types of IDT used recently on cells and a brief perspective on the future application of SAW techniques on live-cell studies was presented.

**Figure 2 micromachines-13-00030-f002:**
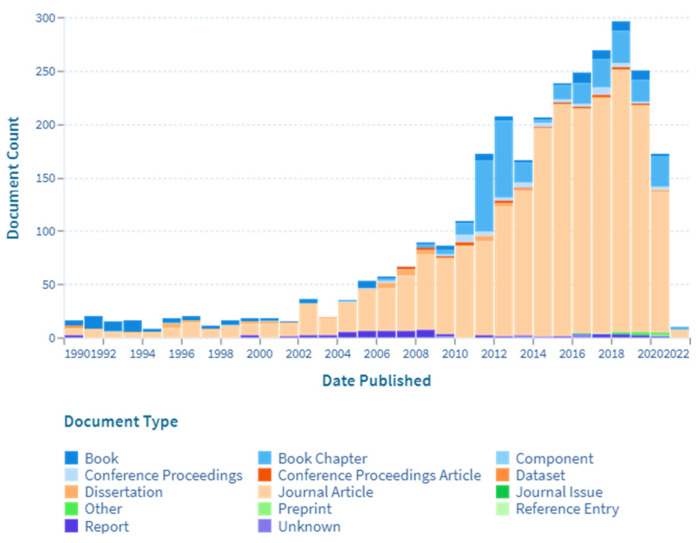
Scholarly work of surface acoustic waves on cellular biology from 1990 based on lens.org [18].

## 2. Application of SAW

Over the last ten years, SAW techniques have been established for cell applications, such as cell manipulation, cell separation, cell seeding, cell migration and proliferation, cell mechanical properties, and cell behaviours. In this section, these applications are summarised and detailed in the aspect of the SAW design properties, main findings, and the sample used in the recent cell studies.

### 2.1. Cell Manipulation

Optical tweezers were previously used by numerous scientists for cell manipulation due to their high precision and adaptability [19]. On the other hand, optical tweezers are bulky and rely on complex and expensive optical setups [20]. Moreover, the associated focussed laser beam can heat the trapping medium [21], while temperatures may cause physiological damage to cells. Therefore, researchers have been developing SAW-based cell manipulation platforms to overcome these limitations [22]. Other advantages of an acoustic cell manipulator or acoustic tweezer include the capacity to manipulate particles and cells in a contactless, label-free, biocompatible, and precise manner [23]. Additionally, unlike hydrodynamic focusing procedures [24], the SAW focusing method does not require high-flow fluid inputs, and does not suffer from particle dispersion.

Figure 3 illustrates the configuration of a SAW device for cell manipulation, in which a pair of IDTs are arranged face-to-face in parallel. When AC voltage is applied, acoustic waves between the two IDT electrodes can be generated. With the generation of SAW, a single or group of cells contained in a fluid is directed to either a pressure node or antinode by the primary acoustic radiation force, and trapped at a specific location. After the cells have been trapped, the SAW device’s frequency and phase can be changed to move them. In this section, cell patterning and cell concentrating were also reviewed, since both techniques are similar in terms of operation and purpose.

Jinjie Shi et al. demonstrated one of the first SAW devices employed in cell manipulation, cells pattern, and other microparticles in a PDMS-based microchannel, as shown in Figure 4a. The input AC signal was tuned to 12.6 MHz, which is the excitation frequency for LiNbO_3_ SAWs in the Rayleigh mode, with an input power of 15–22 dBm. Larger acoustic radiation forces arose from increased power, leading to near cell aggregations. Later, Jinjie Shi et al. utilised polystyrene beads, red blood cells (RBCs), and *E. coli* cells with different diameters to test the device adaptability using the same SAW device reported previously. The researchers also discovered that the needed acoustic tweezers power intensity of 2000 W m^−2^ was approximately 5 × 10^5^ times lower than that of optical tweezers (109 Wm^−2^) [25].

Meanwhile, Lothar Schmid et al. employed the SAW device to pump biologically relevant chemicals, such as red blood cell solution, at a physiological rate of 60 beats per minute in 300 mOsm phosphate-buffered salines [32]. The device was fully integrated on a chip, and did not require any external fluid tubing to prevent the sample from contamination. The device had a finger spacing of approximately 13 µm and worked at 142 MHz. Additionally, the researchers examined the flow vector of blood pumped in different types of channels.

In contrast to previous studies which employed SAW using conventional IDT, Xiaoyun Ding et al. [26] used chirped IDT with a spacing gap that grew linearly from 25 to 50 µm by a 1 µm increment, to generate SAW at a workable frequency range of 18.5 to 37 MHz. By altering the input RF, they can change the location of the pressure nodes created by standing SAW interference, as illustrated in Figure 4b. They discovered that the manipulation of the 10 µm diameter of polystyrene beads required just approximately 0.5 nWm^−2^ for particle velocities of around 30 m/s, which was significantly less than their optical equivalents (106 times less than optical tweezers and 100 times less than optoelectronic tweezers) [33]. The authors also tested the viability and proliferation of HeLa cells under the high-power exposure of SAW up to 23 dBm for 6 s, 1 min, and 10 min. Consequently, after 10 min in the SAW field, no substantial physiological impairment to cell viability or proliferation was observed.

Cell–cell interactions at the single-cell level have also been investigated using SAW methods. Feng Guo et al. [27] used a SAW device to change the spacing and spatial configurations of a few suspended cells at frequencies of 13.35 and 13.45 MHz, and observed the transmission of fluorescent dye between cells to quantify gap junctional intercellular communication in different homotypic and heterotypic populations (Figure 4c). 

In another project by the same team, Kejie Chen et al. [28] employed the orthogonal IDT, where two pairs of IDTs perpendicular to each other were employed to generate acoustic waves for fabricating HepG2 spheroids, as shown in Figure 4d. The wavelength of the IDT was designed to be approximately 150 μm, so that a maximal spheroid diameter of approximately 100 μm could be achieved. Using the SAW technique, more spheroids were produced compared to other techniques [34,35,36]. Thereafter, the drug susceptibility of HepG2 cells under monolayer and spheroid growth created by the SAW device was compared using the standard anticancer agent 5-fluorouracil. Consequently, they discovered that spheroids had higher cell viability than 2D cultures, because the tested drugs spent a longer duration to diffuse before entering the spheroids.

While many researchers manipulate cells within culture medium, Shahid M Naseer et al. applied SAW using a slanted finger IDT on the LiNbO_3_ substrate (Figure 4e) on cardiac cells within gelatin methacryloyl (GelMA). An ultraviolet (UV) light was applied after the cell alignment to cure the GelMA hydrogels. As a result, the cells began to beat after 5 to 7 days of the curing process [29]. Byunjung Kang et al. [30] fabricated therapeutic vascular tissue containing a 3D collateral distribution of vessels using a SAW device at a frequency of 13.928 MHz. The system was used to align human umbilical vein endothelial cells and human adipose stem cells into a 3D cylindroid array of a biodegradable catechol-conjugated hyaluronic acid hydrogel by the combination of SAW and bulk acoustic wave (BAW). The engineered 3D vascular was transplanted into the dorsal regions of mice, as shown in Figure 4f. The therapeutic effects of the fabricated vascular constructions were proven by demonstrating significant tissue recovery upon employing an ischaemic animal model.

Zhenhua Tian et al. [23] used a different type of IDT, which is a spiral-type shape, as shown in Figure 4g, which operated in various resonant frequencies of 10.8, 12.1, 13.9, 20.1, and 23.3 MHz. The device is capable of promoting dynamic and programmable cell manipulation by reshaping SAW wavefields to produce various pressure distributions. Other than the 2D rectangular lattice, more lattice configurations can be generated using modulated excitation pulses with additional frequency combinations, and the SAW direction can be altered using the spiral-shape SAW device.

Recently, Takumi Inui et al. [31] presented a novel approach for removing cells from an adherent cell culture (C2C12 cells) by utilising a 100.4 MHz SAW device with a focused IDT, and achieved a 90% success rate. As shown in Figure 4h, a SAW device integrated with a Petri dishwater gap of 1.5 mm, a PBS immersion period of 300 s, and a 75 V input voltage resulted in the elimination of cells from an area of 6 × 10^3^ mm^2^, comparable to approximately 12 cells. The focused SAW was designed based on Equation (1), where the focal diameter, D, is defined as the region within 4 dB of maximum amplitude.
(1)$$D=\frac{\lambda d_{f}}{W_{n}}$$
where λ is the wavelength, d_f_ is the focal distance, and W_n_ is the aperture of the focused SAW. These studies demonstrated that SAW devices provide excellent platforms for manipulating living cells, indicating their potential in developing future therapeutics. By adjusting the IDT design, SAW input power and amplitude, the living cells can be manipulated to the desired pattern.

### 2.2. Cell Separation

In the previous section, the application of the SAW device in cell manipulation was reviewed. This section proceeds by reviewing cell separation, including cell focussing and cell sorting. Separating cells is an important part of the research process on cell characteristics, disease diagnoses, and treatments. It allows for the label-free, contactless, and biocompatible separation of cells, based on their size and physical features. Cell separation, like cell manipulation, is primarily based on the nodes and antinodes formed by standing waves [37]. Unlike cell manipulation, a standing acoustic field within one stream channel is usually produced. If there is a standing acoustic field in a fluid medium, the cells are driven to pressure nodes at the lowest acoustic radiation pressure, as seen in Figure 5. Cells of different sizes and/or physical qualities are exposed to varying acoustic radiation forces, and moved to the pressure nodes at different periods, resulting in distinct identifiers. The efficiency and speed of this separation technique can be tweaked by adjusting the applied SAW power, SAW working wavelength, flow rate, and channel geometry [38].

Jinjie Shi et al. [38] employed a SAW device to continuously separate dissimilar cells in a microfluidic channel under laminar flow, as shown in Figure 6a. When an AC voltage with a signal frequency of 12.6 MHz is applied to the IDTs, the SAW field propagated in opposite directions, causing pressure variations inside the liquids by generating longitudinal leakage waves. The cells were subjected to lateral acoustic radiation forces as a result of the pressure changes. The larger particles (4.17 µm diameter) moved faster to the centre outlet since they tended to experience larger primary acoustic radiation force, whereas the smaller particles (0.87 µm diameter) remained in the side streams. A power of 15 to 22 dBm or 30–160 mW and a flow rate of 0.6 to 2 mL/min were applied to achieve higher separation efficiency.

Thomas Franke et al. [39] used a SAW device with a unidirectional [29] shape IDT, where the width decreased from 23 to 28 µm. They were successful in demonstrating cell sorting of three different cell types: HaCaT cells, murine fibroblasts L929 cells, and MV3 melanoma cells using a slanted IDT at a lower frequency. However, the sorting was just below 100% at higher frequencies. Figure 6b shows MV3 cells entering the flow from the top to the device targeted area, where three cells are close to each other. To attain a sufficient distance between the cells, the hydrodynamic flow field was accelerated and consequently, the cells successfully moved through different outlet channels (right and left outlet) at a high frequency.

In addition, Thomas Franke et al. and Lothar Schmid et al. sorted cells directly from the medium (OptiPrep™ Density Gradient Medium, Merck, Darmstadt, Germany) without prior encapsulation into drops using the same tapered shape IDT. They tried a different type of cell, mouse melanoma cells, with fluorescence and cell sorting was achieved with the application of a SAW device corresponding to a resonance frequency ranging from 161 to 171 MHz. The device was also capable of rapidly separating cells and other objects, regardless of their sizes or contrast [44].

On the other hand, Xiaoyun Ding et al. proposed a novel configuration of tilted angle standing surface acoustic waves (taSSAW), which were positioned in the microfluidic channel at an angle to the fluid flow direction [40]. Using a SAW device with an input working frequency of approximately 19.4 MHz and input power of 25 dBm, the researchers investigated the effect of the SAW field on different sizes of particles. The biggest size of particles lies at the centre of the microchannel, since the radiation force on the particle was stronger as compared to the smaller particles. It was also observed that the particle trajectories were dependent on acoustic power. For the validation, they successfully divided MCF-7 cancer cells from normal leukocytes (WBCs). The SAW pressure nodal lines were fixed at 15 degrees from the fluid flow direction, as shown in Figure 6c.

In the same group, Peng Li et al. reported an optimised acoustic separation testing platform that improved cancer cell separation throughput by up to 20 times, as compared to what was previously accomplished with the same SAW device, while also increasing separation efficacy. The researchers rearranged the IDT electrode to be titled at 5° from flow direction, and successfully demonstrated rare-cell separation with MCF-7 and HeLa cells, as shown in Figure 6d. Furthermore, various cancer cell lines such as melanoma, and prostate cancer cell types were used to evaluate the high-throughput taSSAW device [41].

Subsequently, the same group designed and tested an acoustic-based circulating tumour cell (CTC) separation system with high throughput. The device includes characteristics such as a hybrid PDMS–glass channel for increased acoustic energy density, and a divider design for a localised decrease in cell flow velocity. They were able to isolate cancer cells from a mixed solution of WBCs at a flow rate of 7.5 mL/h, and they also performed CTC isolation and phenotypic characterisation from clinical samples from male patients with metastatic prostate cancer [42]. As shown in Figure 6e, when the acoustic field is triggered, PC3 cells are driven toward the bottom collecting outlet, while most WBCs continue to flow to the waste exit. Previously, Mengxi Wu et al. successfully separated exosomes from blood cells using two pairs of SAW devices, as shown in Figure 6f. The first pair of SAW devices were used to remove red blood cells (RBCs), white blood cells (WBCs), and platelets (PLTs), while the second pair separates the exosomes from apoptotic bodies (Abs) and microvesicles (MVs) [43]. In summary, it was established that SAW devices provided high efficiency in separating live cells based on the size of cells. The major factor influencing the efficiency of sorting is the fluid flow rate and the configuration of IDTs.

### 2.3. Cell Seeding

Acoustic waves can be employed to drive particles to particular regions to form the scaffolding for cell growth, as illustrated in Figure 7. In vitro cell culture is a technique for cultivating cells in extracellular matrices and possibly producing artificial organs in a laboratory setting. Cell proliferation requires an even distribution of seed cells in the matrix scaffold, as well as the efficacy of the seeding process. Moving a cell suspension into a polymer scaffold material takes hours or days without external driving forces. Moreover, as stated previously, the SAW device can be used to move droplets with high internal streaming and agitation into polymer matrices for cell growth [16].

The important parameters influencing the efficiency of the SAW device for cell seeding are input power, using higher power. However, strong SAW radiation may cause the atomisation of the droplet carrying the suspension and disrupt cells’ ability to penetrate the targeted scaffold [44]. Furthermore, increasing input power only slightly improved seeding efficiency, and in the worst case, the polymeric scaffold material melted and absorbed the SAW radiation [45].

Haiyan Li et al. [46] set up an experiment for cell seeding using SAW actuation, as shown in Figure 8a. They showed that the SAW-driven seeding process with RF power of 570 mW took about 10 s; substantially faster than gravity-driven diffusional processes (which took more than 30 min). They also showed that the SAW approach could force particles deeper into the scaffold, resulting in much-improved particle distribution uniformity. To validate that the SAW is effective in seeding live cells into scaffolds, the researchers investigated how the SAW affected cell viability, proliferation, and differentiation. As a result, they discovered that, after being treated with SAW at 20 MHz for 10 to 30 s with an applied power of 380 mW over a wide range of cell suspension volumes (10–100 µL) and cell densities (1000–8000 cells/L), more than 80% of the primary osteoblast-like cells were found to be viable [47], as shown in Figure 8b.

An investigation of the influence of SAW-based agitation on the efficiency of suspended fluorescent particles in a polycaprolactone (PCL) scaffold was performed by Melanie Bok et al. [45]. In just a few seconds, the researchers demonstrated that SAW devices were capable of setting the particles throughout the polymer matrix in equal intervals of efficiency of up to 90%, with RF power of 400 mW and a frequency of 20 MHz. Additionally, they verified that the SAW radiation treatment did not adversely affect the cells and the ability of the cells to proliferate.

Sixing Li et al. [48] established an alternate seed and cultivation platform for two cell lines using a SAW device, as shown in Figure 8c. First, one type of cell was modelled with SAWs on the surface of the LiNbO_3_ substrate, and the second cell type was moved into the node lines of the first type by a phase shift of the SAW field. Co-culture of the cells HeLa and HMVEC-d led to the discovery of the improved mobility of cancer cells. This could be initiated by cross-talking begun by endothelium cells that improved cancer cells by employing STAT3/Akt/ERK, a5b1 integrin, and GTPases.

The water coupling layer between the IDTs and PEGDA-GelMA polymer-based tube was placed by James P. Lata et al. onto a LiNbO_3_ substrate (Figure 8d). This configuration was used for the propagation of leaked waves in the tube with the setup input power density of 1.5 Wcm^−2^, and resonant frequency of 12.65 MHz of the SAW device. As a result, both 10 μm beads and HeLa cells were successfully seeded into the polymer-based hydrogel, to simulate the cellular structures that provided further physiological tissue function [49]. 

In summary, the cell seeding using the SAW device was led by James Friends and colleagues by successfully seeding cells in a 3D tissue scaffold in a very short time (~10 s) for 2.8 × 104 fluorescent particles in the 20 µL droplet. The researchers also showed that SAW irradiation does not result in cell denaturation or negative effects on the cells’ ability to develop and proliferate. These findings inculcate confidence in the SAW as a viable and potentially versatile and powerful method for seeding cells into scaffolds rapidly, efficiently, and uniformly.

### 2.4. Cell Migration and Proliferation

Cell migration is crucial in physiological and pathological processes, such as wound healing, angiogenesis, morphogenesis, and cancer metastasis [50,51]. Although the majority of studies use chemical variables, it was reported that mechanical stimulation can dramatically control cell migration by electrical cue [52,53], substrate stiffness [54,55], stretching [56], and acoustic waves [57]. Most of these simulations are based on invasive procedures, except electrical and acoustic waves cues [58,59]. Cell migration is the recent cell application by utilising the SAW technique. However, there is little research regarding SAW devices on cell migration, and the systematic exploration of these devices is presently lacking. 

Stamp et al. [57] fabricated a SAW chip with the periodicity of the IDT electrode at 25 µm with 159 MHz of the resonant frequency. They applied the SAW field with input power from 3 to 6 dBm to adherent Saos-2 cells in a wound-healing assay. The setting of the excitation wavelength was according to the cell dimension, while the applied power was comparable to the reported ultrasound treatments. Consequently, the adherent cells in the SAW sound path had favourable effects on their migration behaviours, as the artificial wound recovered up to 17% higher than control samples, as shown in Figure 9a. They suggested that SAW directly promotes cell migration and development through dynamic mechanical and electric stimulation.

Jonathan Rosenblum et al. [60] used a commercial SAW device manufactured by NanoVibronix Inc. (Elmsford, NY, USA). They successfully increased the epidermal thickness in skin explants as compared to the untreated controls under the commercial SAW device, as shown in Figure 9b. The researchers observed that ultrasonic stimulation increases activity in the base layer of the epidermis, as shown by the increase of the CK14+ in this layer, with a low acoustic wave frequency of 89 kHz, and an application power of 0.4 mW/cm^2^. The increase in CK14 was expected to contribute to the improvement of epidermal tissue, since it was expressed in basal epithelial cells [66], which were exposed by acoustic waves and important precursors for indicating cellular division.

Gina Greco et al. [61] fabricated a SAW chip with frequency at approximately 50 MHz and 20 μm wide fingers. They tested the viability, proliferation, and morphological analysis of the cell line U-937 with RF inputs of 21.5 dBm. Using a SAW device with a higher amplitude, they found an enhanced proliferation of approximately 36% with respect to the control static conditions. Data showed that the high recirculation of the fluid within the Petri plates may be induced while negligible heat is maintained. Furthermore, there was no increase in cell death and the cell shape remained constant when SAWs were present. The authors hypothesized that the heating caused by continuous-wave operations from SAW may be important for optimal cell cultures. Besides, they found that fluid shear stress induced by the radiation force of the SAW device can induce positive effects in cells. This finding contradicts the reports from a previous study, where the fluid recirculation led to fluid shear stress in the Petri plate (<10 mN/m^2^) applied to the cells, and subsequently damaging the cells [14]. The cell proliferation test results on U937 cells after 48 h are shown in Figure 9c. SAW1 and SAW2 demonstrated different fluid shear stress at 120 mN/m^2^ and 280 mN/m^2^, respectively.

While Yangcheng Wang et al. [62] used a SAW device differently by producing a biocompatible hydrogel substratum that employed SAW-field with micro wavy and reticular-patterned microstructures to increase cell adhesion and migration. They reported that the viability and proliferation of L929 mouse fibroblast cells enhanced the surface of polymer-based patterned microstructures fabricated by a SAW device instead of using the physical mould. The SAW field was generated on pre-polymer liquid film to produce the micro wavy patterned, and UV light was used to harden the patterned surface.

Brugger et al. [63] reported a new field of application of SAW-based sensors using the LiTaO_3_ 36° XY-cut piezoelectric material with the thin layer of SiO2-cover, where cell growth was observed with the horizontal shear acoustic surface waves (SAW) (Figure 9d) at 208 MHz. The characteristics of cell growth in the confluent cells layer could be examined without microscopic observation by studying the phase-shifting signal changes. Furthermore, the researchers employed two different piezoelectric substrate materials and the direction of SAW propagation: Raleigh waves were excited on LiNbO_3_ 128° Y-cut in the main direction and Love waves were generated on LiTaO_3_ 40° XY-cut (Figure 9e). They found that Raleigh waves influenced the cell migration on the closure of an artificial wound in a cell monolayer, in a distinct power range. Upon stimulating the ectodermal cell line MDCK-II with of SAW, a significant improvement of cell growth and migration rate up to 135 ± 85% at an input power of 4 and 8 mW was observed. On the other hand, the cells detached either from the substrate surface or became necrotic within minutes to hours at power levels of 128 mW [64]. However, Love waves were not effective in eliciting cell migration and proliferation, since they were horizontally polarised horizontally, thereby missing substrate vibration as they hypothesised that mechanical stress may affect the cell behaviours.

Recently, Imashiro et al. developed a SAW device to apply acoustic waves to cells on the substrate with a centre frequency of 14 MHz as shown in Figure 9f. They eliminated the effect of electric field on cell migration due to the application of a glass substrate which is non-conductive material. Surprisingly, they found that the migration rate of 3T3 fibroblasts was increased by 42% under the exposure of acoustic waves for 8 h observation [65].

The enhancement of cell migration may be contributed by the stimulation of dynamic pressure on adherent cells induced by the SAW device [57]. SAW generated-electrical stimulation has also been highlighted as a factor that improves wound healing. However, it was confirmed that mechanical stimulation is a dominant factor of faster cell migration upon eliminating the electric field using conductive layers on the piezoelectric. Other factors, such as fluid shear forces and temperature, could be neglected when low power is applied to the IDT electrode. The movement of the substrate by the SAW may affect the cellular molecular level, since it was previously reported that the cell membrane deformation or stretching increased the calcium concentration, Ca2+ level [67,68], thus it may contribute to the faster cell migration. Despite the uncertainty of the underlying mechanism that promotes cell migration using the SAW device, these studies revealed that the SAW device can be utilised to replace drug biochemicals to solve the issues in the wound healing process.

### 2.5. Cell Mechanical Properties

Cell mechanical property analysis is useful for both diagnostic and cell biology studies. Mechanical property changes in cells are commonly associated with many diseases such as cancer and malaria, as well as atherosclerosis and hypertension [69]. Cell mechanical properties are represented by several biomechanical parameters, such as deformability, Young’s modulus, and compressibility. Several methods are available to measure the mechanical properties of cells, including microfluidics [70], micropipette aspiration [71], a combination of microfluidic and micropipette, micro-post arrays [72], magnetic twisting cytometry [73], optical stretcher [74], parallel plate [75], and acoustically actuate bubble-based technique [76], and atomic force microscopy (AFM) [77]. Although these approaches offer great resolution in the measurement of displacement and force, most of these methods have low-performance levels and low throughputs. These methods are therefore not suitable for many other applications like clinical diagnoses; however, some fundamental biological trials may be appropriate. Due to their advantages in high performance and non-invasive measurement technology, SAW devices provide an alternate method for cell mechanotyping.

Sukru Ufuk Senveli et al. [78] trapped a single cell in a microcavity placed at the centre between a pair of IDTs on ST cut quartz substrate, as shown in Figure 10a. The researchers applied SAW at a passband peak frequency of approximately 200 MHz to observe the elastic moduli of a single adherent cell. As a result, they found that the elastic modules of SKBR3 and MCF7 cell lines were not distinct. MDA-MB-231, on the other hand, revealed a significant difference with an approximate stiffness of 2.516 ± 0.054 GPa from any other cell line.

A SAW device was fabricated by Yanqi Wu et al. [79] by employing a microfluid channel for measuring cell compressibility and distinguishing cell mechanophenotype in a suspension condition. The width of IDT electrode fingers was 300 μm, indicating that the frequency of resonance is 12.8 MHz. Figure 10b illustrates the working principles of the SAW device. By changing the phase of one of the IDTS, the targeted cell moves to a new nodal line. This transition is recorded and applied in the acoustic force radiation, Stokes drag force, and an equation related to compressible sphere proposed by Doinikov [82] to measure cell compressibility. For the cell compressibility and cell size tests, three human alveolar basal epithelial cells (A549 cells), human cell smooth cells (HASM cells), and MCF-7 breast cancer cells were used. The device successfully distinguished cell types based on their physical characteristics (density and size), along with the cell’s capability to compress.

Xinwei Wei et al. [80] used IDTs electrodes consisting of 50 split-finger pairs with a finger’s width of 7 μm constructed on a piezoelectric quartz substrate, as illustrated in Figure 10c. The IDTs based on Ti/Au (20/200 nm) were chosen as the electrode metal type to generate shear horizontal acoustic, also known as Love Wave. This device was used to study the contractile properties of HL-1 cardiomyocytes, where variations in the insertion loss and phase position of the sensor reflect mechanical changes in contraction and viscoelasticity.

Recently, Link et al. demonstrated the generation of an acoustic field using a single IDT printed on a LiNbO_3_ substrate with 60 finger pairs of electrodes, and operating at 162.2 MHz to measure both viscous and elastic modulus of single red blood cells (RBCs). 

According to Figure 10d, the generated SAW interacts with the channel wall and resulted in a knife-edge effect that excited a cylindrical wave at the SAW’s entry point. The interference between the cylindrical wave and the SAW propagating on the substrate produces an acoustic pattern with spacing λnf, which is a critical characteristic for determining the viscous and elastic properties of erythrocytes [81]. Like Yanqi Wu et al., the researchers revealed the viscous and elastic properties of an individual RBC by using the equation of acoustic radiation force on a spherical object with volume. In summary, the frequency of SAW devices may differ according to the method and configuration of measuring the cell mechanical properties of the sample. Therefore, researchers must consider an appropriate SAW device to be applied on cells to provide a high-throughput method for cell mechanotyping trials.

### 2.6. Cell Behaviours

SAW technique may cause different cell behaviours. For example, the acoustic waves may alter the physiological, molecular, and biochemical changes induced in living cells [83]. Therefore, a few researchers investigated the effect of SAW on cell behaviours in different setups of acoustic power and frequency. Besides, SAW devices have also been used to detect the different behaviours of cells. However, the associated bio-effects on cells under acoustic waves are not clearly understood.

A range of SAW devices with LiTaO_3_ substrates (Figure 11a) was developed by Z. Racz et al., which operated at various frequencies, ranging from 60 to 868 MHz to detect biomolecular agents and micromolar octopamine [84]. The octopamine is a type of receptor-specific ligand binding that is important in determining insect behaviours [85]. SF9 insect cell line was used as a functional layer for the SAW device, since the cells endogenously express the octopamine receptor. Due to ligand-receptor interaction, the SAW biosensor is able to detect octopamine reactions inside cells. This depends on the SAW penetration depth, which, in turn, is defined by differential frequency change, since the 12.5 μM octopamine is measured at approximately 590 Hz.

Tao Wang et al. demonstrated that shear horizontal-surface acoustic waves (SHSAW) may be utilised to evaluate cell mass loading in two-dimensional suspension cultures, as well as three-dimensional cell culture models for detecting and quantifying cell growth changes over time [86]. It comprises two pairs of resonators consisting of IDTs and reflecting fingers, as shown in Figure 11b. The researchers coated a ZnO layer on piezoelectric substrates LiTaO_3_ to decrease the coefficient of temperature and increase the sensitivity to mass [89]. The SAW device was tested at 14.05 MHz at different cell concentrations with operating frequencies. The result showed that the frequency response was relative to various cell concentrations.

On the other hand, Citsabehsan et al. [87] demonstrated differences in cell behaviour in terms of proliferation, membrane permeability, metabolism rates, cell adhesion and morphology, in response to acoustic stimulation. They used a SAW device with an operational frequency of 48.5 MHz and applied the acoustic waves to the cells inside the curved microchannel, as shown in Figure 11c. The curved microchannel was used to facilitate prolonged exposure times of SAW by stimulating several cell lines such as HaCaT (human keratinocytes), L929 (mouse fibroblast), MSCs (mesenchymal stromal cells), and MG63 (mesenchymal stromal cells) (mouse osteosarcoma). As a result, they showed that acoustic exposure can impede cell attachment, diminish cell spreading, and, most intriguingly, boost cellular metabolic activity without affecting survival rates.

Cryopreservation is essential for extending the shelf life of delicate biomaterials while preserving cell functions. To suppress intracellular ice formation during freezing, a cryoprotective agent (CPA) is required, and it must be removed prior to clinical use due to its toxicity. Using a SAW device, Umar Farooq et al. [88] demonstrated a new approach of multistep CPA loading/unloading that was quick, and could considerably increase post-cryopreservation cell viability. Moreover, the technique improved the previous multistep CPA loading/unloading device which consumed more time, and often caused osmotic shocks and mechanical injuries for biological samples. For the test, an RF capacity of 28 dBm was applied to the SAW (Figure 11d) at 19.87 MHz, followed by a droplet of 30 μL on the surface near the IDT to generate a circulating pattern for the mixing process. In summary, the application of SAW in sensing and evaluating the cell behaviours provided fast, rapid and high-throughput testing. Hence, most studies employed Love waves as compared to Raleigh waves, due to their high sensitivity and compatibility for sensing in liquid conditions.

## 3. Types of IDT Electrodes Used in SAW Techniques for Live Cell Studies

IDTs are one of the most often utilised transducers for many technical and analysis applications, due to their low cost, ease of manufacturing, and good sensitivity [90]. Generally, the SAW is generated by the combined effect of tension and compression of a piezoelectric substrate at the surface after applying an electric potential to the IDT electrode. The IDT design determines the resonant frequency, f_o_, of the SAW device by the finger’s geometry, which is equivalent to one-fourth of wavelength, λ and it also depends on the acoustic velocity of a substrate, Vs. The relationship between the resonant frequency, f_o_, wavelength, λ, and acoustic velocity of a substrate, V_s_ is expressed in Equation (2).
(2)$$f_{o}=\frac{V_{s}}{\lambda }$$

In addition, the response frequency, direction of propagation, and bandwidth of the SAW generated are determined by other IDT structural characteristics such as electrode shape, aperture, wavelength thickness film, and an electrode pair number [91].

To date, there are several designs and classifications of IDT that have been invented to achieve its functions and applications. The following section discusses the types of IDT designs that have resulted in the development of multidirectional SAW transducers on a single piezo-electric substrate, that could have the advantage of selectively utilising acoustic wave-specific features that are propagated along a given crystallographic.

### 3.1. Conventional IDT

The first and most simple IDT [8] is composed of straight, rectangular fingers which lie on a piezoelectric substrate surface and are connected to contact pads at each end, as illustrated in Figure 12a. This structure creates a range of electrical fields that alternate direction from transducer finger pairs, and in turn generate the inverse piezoelectrical effect to alternate areas of compressive and tensile stress within the substrate. This causes every pair of fingers to displace the substrate, oscillate, and radiate a SAW. The typical IDT is easy to build using a standard photolithographic technique, due to its simplicity and relatively wide strip width.

### 3.2. Tilted Angle IDT

For the titled angle, IDT configuration using a pair of IDT electrodes, the pressure nodal lines induced by the standing SAW are inclined in a particular angle towards the flow line, instead of being in parallel. Peng Li et al. demonstrated their improved titled-angle SSAW (taSSAW) was able to reach high-throughput separation by having smaller tilted angles with less power input. This result may be because smaller inclined angles allow for longer travel durations between pressure antinodes and nodes during separation. Thus, the taSSAW device overcomes the constraint of current acoustic separation techniques, which limit the total separation distance to a quarter of the wavelength of the acoustic wave [41].

### 3.3. Single-Phase Unidirectional Transducer (SPUDT) 

SPUDT, based on double-electrode IDT (Figure 12b), has a relatively narrow strip width (~λ/8). This type of IDT is frequently used when the frequency response needs to be controlled precisely. This is because a single electrode of IDT generates waves with equal amplitudes in both directions that are capable of drastically deteriorating the performance of SAW devices [92]. Therefore, the double electrode used in SPUDT is designed to solve the issue by decreasing the reflection loss from a source. SPUDT has been used by Layla Alhasan and others to generate spheroids that were loaded on the SPUDT device with a medium of human breast cells (BT474). An electric signal oscillating at resonant frequency was delivered to the SPUDT to generate a surface acoustic stream. The researchers demonstrated that the spheroids can be obtained in less than 1 min in uniform form and dimensions [93].

### 3.4. Slanted-Fingers IDT

The slanted-finger IDT (SFIT) (Figure 12c) was formerly utilised as a mid-band and wide-band filter in data terminals [94], however, recently, it has been widely used in the biomedical field, especially to handle droplets and particles. Different wave profiles can be obtained by setting the input frequency via SFIT, also known as the tunable SAW device. The resonant frequency was different from one end to another depending on the finger’s width. For instance, Yannyk Bourquin et al. manufactured SFITs of 250 μm to 500 μm wavelength to control the lateral position of the excitation wave, ranging from 16 MHz to 8 MHz [95] frequencies, to separate particles by their particle or cell size.

### 3.5. Focused IDT

Kharusi and Farnell [96] debated focused IDT (FIDT) for the first time in 1972. They proposed two different shapes of FIDTs: circular arc as shown in Figure 12d, and wave surface. The results indicated that both types of FIDTs were capable of focusing the SAW, whereas the FIDT with the wave surface shape exhibited better-focusing properties. Basically, FIDT is achieved by altering the electrode design in the acoustic energy of the SAW device from linear to circular focal points [97]. The FIDTs acoustic waves focus directly on a localised area, and the focusing acoustic energy maximises the gradient of acoustic force [98]. This leads to the production of waves of higher amplitude, and could result in higher streaming speeds.

Other advantages of FIDT can be generated by concentrating the SAW energy on the centre of the IDT with higher intensity, a high beam-width compression ratio, and a small, localised area. The FIDT can achieve the same mixing efficiency with a far lower SAW power compared with the conventional IDT [10]. The FIDT was used by Collins et al. to create a range of 25 μm for sorting 2 μm-diameter particles at frequencies of 386 MHz. Their study showed that the FIDT system improved streaming efficiency compared with conventional SAWs with acoustic energy focusing and acoustic streaming [99].

### 3.6. Chirped IDT

The chirped IDT (Figure 12e) has a linear gradient in the finger distance that allows the SAW to resonate on a wide range of frequencies by tuning the input signal. Contrary to the standard IDT design, in which all of the fingers and spaces are identical in width, an IDT chirped with increasing finger and spacing width (λ) gradually. The chirped IDT constructs a finger with its spacing width of half a period in the wave. The bandwidth of the SAW device can be increased by varying finger-width to provide a broad spectrum of available SAW wavelengths, thus resulting in a wide range of manipulation or sensing. Therefore, the device with chirped IDT thus has a higher independent frequency than the device with standard IDTs [100].

### 3.7. Multiple IDT

The multiple IDT configuration has been explored, such as spiral IDT [22], orthogonal IDT [26], and hexagonal IDT, as illustrated in Figure 12f [101]. The image consists of two different and aligned delay paths, so that acoustic waves can be generated in many directions by properly designing excitation signals. As SAWs can be controlled independently in more directions, further wave patterns are generated, and particles/cells can therefore be designed in additional configurations.

This design also allows multiple properties of a thin-film material to be simultaneously extracted to achieve more complete characterisation than when using a single SAW device. In principle, this would enable the fabrication of a multiple SAW system capable of operating in both gas and liquid phases, as well as facilitating the use of the same device for chemical and biosensing applications according to the crystallographic directions identified for a particular piezoelectric substrate. However, multiple IDT settings may need to be operated by multi-channel functions and high-end programmable electronics.

All type of IDT-based SAW device has been designed to fulfil the specific applica-tion to live cell in vitro studies. Table 1 summarizes the advantages and limitations of the various type of IDT-based SAW devices.

## 4. Perspectives

A combination of SAW device and microfluidic channel (acoustofluidic) is suitable to be implemented in live-cell studies, especially in cell manipulation, cell sorting, and cell migration. This is due to the feature of the generated acoustic waves providing proper resolutions and wavelengths size, which are compatible with the size of cells, ranging from several microns to 100 µm. Besides, these SAW devices are ideal platforms in cell biology because of their biocompatibility, adaptability, and simplicity.

Despite its rapid expansion and exciting applications, there are a few points to consider for the future development of SAW device-based cell analysis approaches. First, designing the IDT of the SAW device is critical, since a few parameters need to be considered, including a piezoelectric substrate, finger’s width, electrode metal type, aperture, delay line, and several fingers. Many researchers used LiNbO_3_ as the piezoelectric substrate, since it has a higher electromechanical coupling coefficient than LiTaO_3_ and Quartz, thus reducing the noise and providing lower attenuation [104]. To generate a high resonant frequency of SAW, the finger’s lower width needs to be designed, and it depends on the capability of the available photolithography facilities, such as mask patterning and UV exposure. Several types of metal, such as aluminium and gold, can be deposited on a piezoelectric substrate. However, a thin layer of chromium or titanium is needed as an adhesive layer to have better contact between the electrode and substrate [42]. Most researchers utilised an electron-beam evaporation [105] and thermal evaporation technique [106] to deposit the thin film electrode since the deposition thickness can be controlled easily, and it can be worked in a high melting point process.

Second, the selection of SAW frequency is important to be matched with the application on cells. For example, one of the first applications of the SAW device on live-cell studies was cell seeding, where the frequency of SAW was set at approximately several 10 MHz sufficient to move a drop of cell suspension toward the artificial scaffold. The SAW device has also been utilised to manipulate and sort cells using a wave ranging from 10 to 100 MHz, and producing an acoustic wave wavelength of 15 to 150 μm of water to manipulate cells within 20 to 100 µm in diameter [107]. This frequency range is sufficient to trap cells in the microfluidic, considering the attenuation of SAW occurred through different media, such as glass, PDMS, and liquid. Recently, a higher frequency of SAW device up to 400 MHz was utilised, since the focus application of SAW was diverted to cell migration and cell behaviours. Table 2 shows a summary of various SAW devices that have been developed in cell studies, based on the resonant frequency.

Third, the type and configuration of IDT were reported to influence the operation of a SAW device. Therefore, SAW devices have undergone several changes and improvements to suit certain applications. Starting from the single port, SAW to the current multiple and complex IDT, such as spiral IDT [22], has been designed and fabricated based on their applications. For example, to control cell migration, the gap between IDT electrodes is fabricated according to the diameter of the adherent cell. To separate cells in a limited space such as in a microfluidic channel, the IDT is tilted in several angle degrees to generate more pressure nodes, compared to those produced by the IDT that is parallel to the channel. Besides, a single IDT can be designed in different widths of electrodes, for having multiple ranges of frequency to separate cells in different shapes and sizes.

Fourth, one of the most crucial aspects to be considered when applying SAW to cells is the viability of cells during or after the excitation of SAW. Two types of excitations have been reported: continuous waves and pulsed waves. To date, there are no reports of SAW being harmful to cells. However, since energy must be dissipated on a chip to produce fluid and particle motion, long-term exposure to SAW may detach cells from the substrate [9]. Xiaoyun Ding et al. observed that the temperature increased from 25 °C to 27.9 °C after 10 min acoustic power with 23 dBm; nevertheless, no significant physiological damage on cell viability and proliferation was observed in the standing SAW field within the period. 

Lastly, many researchers have been able to successfully regulate cell behaviours, for example, Haiyan Li et al. reported that primary osteoblast-like cells were delivered into the PCL scaffold in seconds under acoustic waves [45]. However, the ability of IDT-based SAW device needs to be proved for clinical practise, especially in developing a multifunctional implantable bone device. Furthermore, a self-powering system is needed to electrically supply the IDTs, for example, hybrid triboelectric-electromagnetic generators is a promising technology [115] to be integrated with SAW device to avoid bulky and complex system to power up the SAW devices. 

## 5. Conclusions

This review presents the application of the SAW device and types of IDT electrodes used in SAW techniques for live-cell studies. SAW had previously been widely used in communication systems, TV applications, radars, satellite, and mobile telephone applications [116]. For example, SAW is used as filters and duplexers in communications systems for precision and sharp signal filtering. Recently, the application of the SAW device has been implemented in cellular biology research. Due to the advancement in visualising microparticles [99], researchers have relentlessly sought to harness acoustic waves as a tool to be used in live-cell studies. As compared to the SAW device used in telecommunication, a very high wave excitation up to several GHz ultrasonic range is employed. However, the far narrower range of 20 kHz to 400 MHz is used in cell manipulation, cell separation, cell migration, and cell properties, as depicted in Figure 13.

For cell manipulation, cell separation, cell behaviours, and cell mechanical properties, the major mechanisms of these processes are mostly based on the pressure nodes in the liquid generated by the SAW. However, it is still unclear if the underlying mechanism affects cell migration. Several parameters, such as acoustic streaming, intercellular protein agitation, heat generation, and electrical stimulation, need to be considered in order to reveal the most significant factor that influences cell migration [57]. To observe the sole contribution of SAW on cells, the acoustic streaming that induces drag forces to cells could be minimised by altering the microchannel geometry [117]. Both electrical and mechanical components may be generated to cells by applying an AC signal to IDT. To reduce the effect of the electrical signal, Brugger et al. applied a different frequency outside the bandwidth of the resonance frequency [64].

Finally, this review is useful to researchers in designing and developing a SAW device for live-cell studies. To develop a SAW device, it must be established in accordance with the desired application, which includes IDT electrode and delay line, manufacturing, and peripheral equipment.

## Figures and Tables

**Figure 1 micromachines-13-00030-f001:**
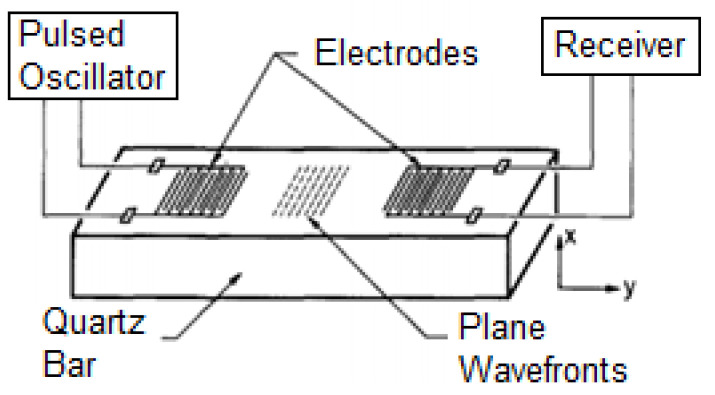
The first illustration of the arrangement for surface wave transduction using IDT on quartz substrate by White and Voltmer. Images reproduced with permission from references [9].

**Figure 3 micromachines-13-00030-f003:**
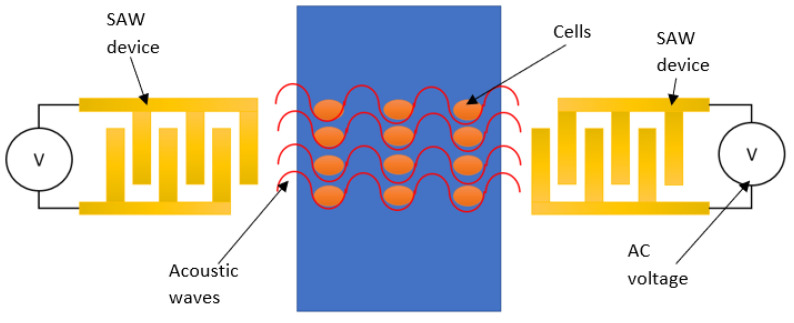
Illustration of cell manipulation using SAW device.

**Figure 4 micromachines-13-00030-f004:**
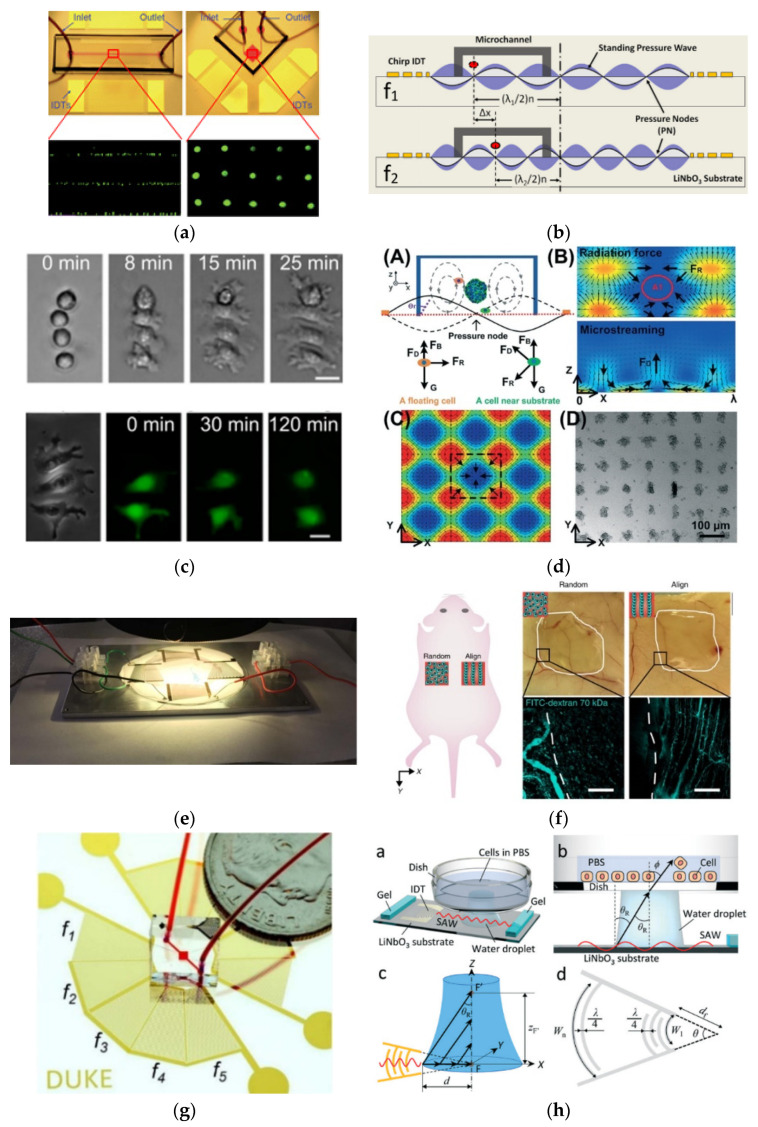
(**a**) SAW devices with a wavelength of 200 µm used in 1D and 2D patterning experiments [25]. (**b**) Driving chirped IDTs at frequencies f1 and f2 produces a standing SAW field [26]. (**c**) Assembly and attachment of HeLa S3 cells and dye transfer between the attached cells [27]. (**d**) Acoustic tweezers were used to produce spheroids above pressure nodes [28]. (**e**) Experiment setup to apply SAW on cells using slanted fingers IDT [29]. (**f**) Comparison between the transplantation of random and aligned cell-hydrogel constructions into the subcutaneous space of the mouse back and the results after 1 week of transplantation [30]. (**g**) A fabricated chip with the spiral shape of IDTs deposited on LiNbO_3_ substrate [23]. (**h**) Locally removing cells from a culture surface by using focused IDT using SAW technique [31]. Images reproduced with permission from references [23,25,26,27,28,29,30,31].

**Figure 5 micromachines-13-00030-f005:**
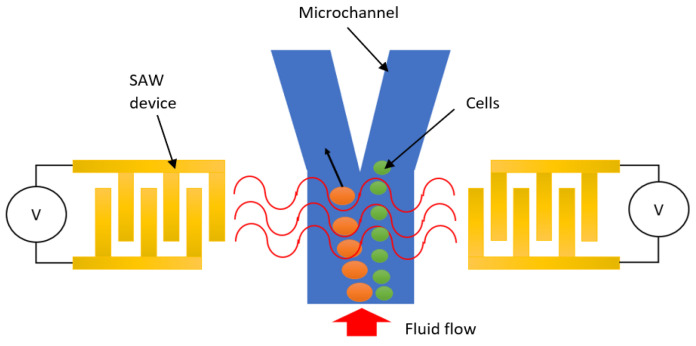
Illustration of cell separation using SAW device.

**Figure 6 micromachines-13-00030-f006:**
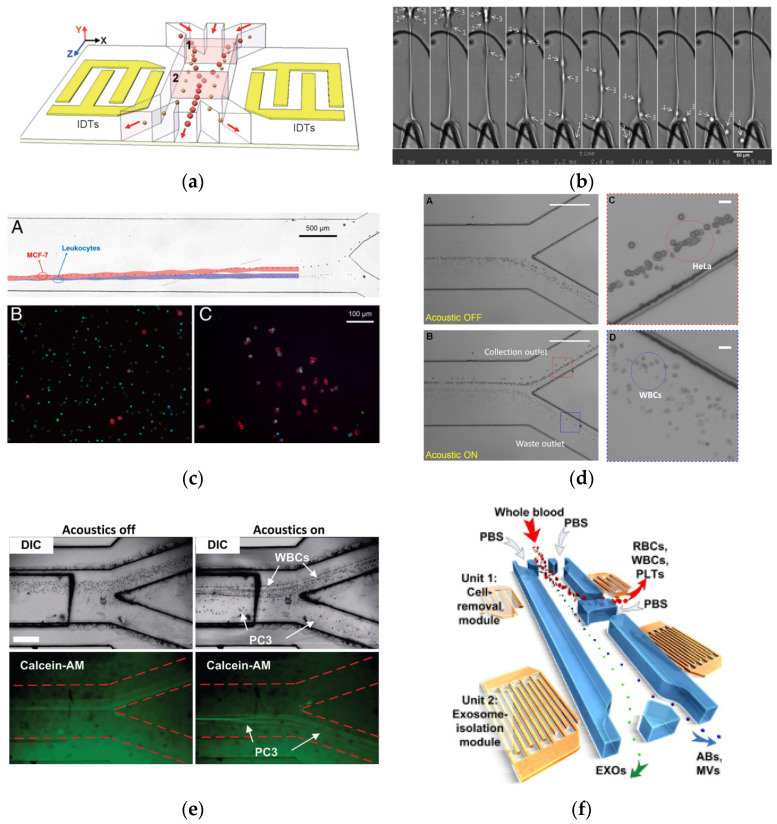
(**a**) Separation process showing larger particles closer to the channel centre and smaller particles further away from the centre [38]. (**b**) MV3 cells enter the different outlet channels at 1 kHz [39]. (**c**) (**A**) Single MCF-7 cell was pulled out from the stream of leukocytes. (**B**) Fluorescent images of cells before separation. (**C**) Fluorescent images of cells after separation [40]. (**d**) (**A**) Acoustic field OFF, no separation was found. (**B**) The acoustic field is ON, larger HeLa cells have been moved to the collection outlet (blue box), whereas smaller WBCs remained in the waster outlet (red box). (**C** and **D**) Zoomed-in images of the collection outlet and the waste outlet, respectively [41]. (**e**) Separation of 5.84 μm beads (not labelled) and 970 nm (labelled with fluorescent dye) using a taSSAW device (Scale bar: 500 μm) [42]. (**f**) PC3 cells were stained by Calcein-AM and mixed into a 1 mL suspension of WBCs. After the acoustic field was activated, PC3 cells were pushed toward the bottom collection outlet (Scale bar: 400 µm) [43]. Images reproduced with permission from references [38,39,40,41,42,43].

**Figure 7 micromachines-13-00030-f007:**
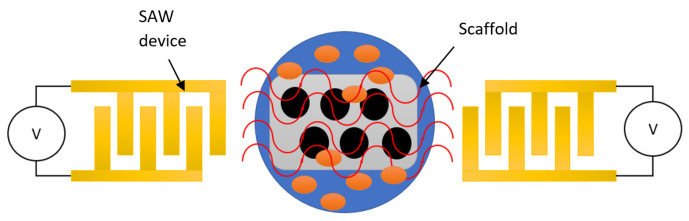
Illustration of cell seeding using SAW device.

**Figure 8 micromachines-13-00030-f008:**
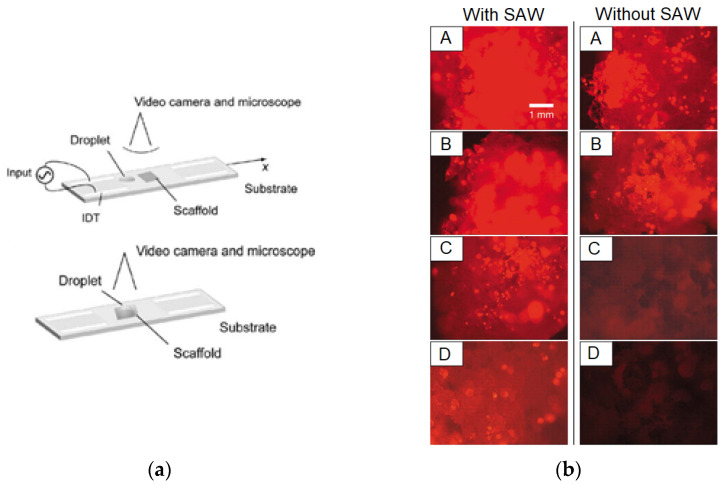
(**a**) Scheme of the experimental cell seeding setup by SAW [46]. (**b**) Comparison between SAW-driven seeding (10 s) and static seeding (30 min) after the cells penetrate the scaffold [47]. (**c**) SAW-based sequential cell seeding with the co-culture results: HMVEC-d cells (green fluorescent) and HeLa cells (red fluorescent) [48]. (**d**) Schematic representation of the whole process in perpendicular orientation for the production of patterned cell fibres using SAW [49]. Images reproduced with permission from references [46,47,48,49].

**Figure 9 micromachines-13-00030-f009:**
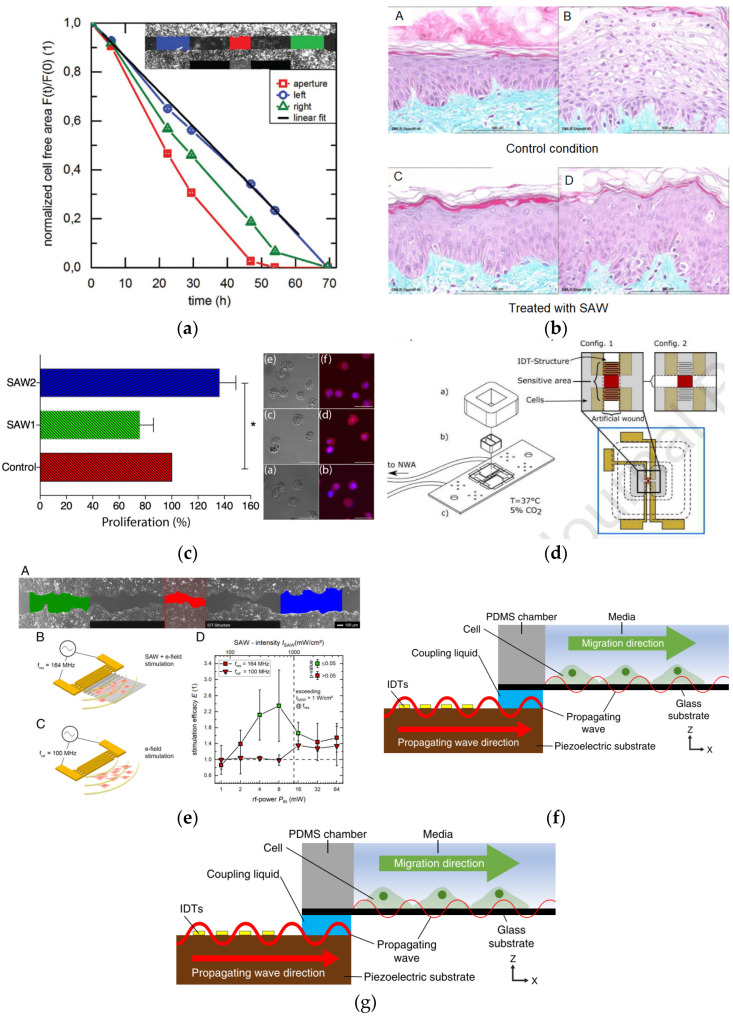
(**a**) The average overall aperture migration speed is 3.8 to 1.2 μm/h, whereas only 3.1 to 1.4 μm/h could be reached from the control sample [57]. (**b**) Comparison between untreated epidermal and treated under continuous acoustic waves showing the epidermal thickness increased by introducing acoustic waves [60]. (**c**) Cells with U-937 seeds cultivated without SAW (control) in the green and blue column, and with the presence of SAW1 and SAW2 in the red column have been tested for proliferation on U937 cells after 48 h in the case of seeding [61]. (**d**) Diagram of the four-step patterning microstructure fabrication procedure [62]. (**e**) View of SAW-main chip components. The delay line area and IDTs in two configurations have been enlarged [63]. (**f**) SAW stimulation of the ectodermal cell line MDCK-II. Power dependency on SAW stimulation at different frequencies. There was a significant improvement in cell growth and migration rate up to 135 ± 85% for SAW [64]. (**g**) The propagating SAW was transferred to the glycerol layer, the glass substrate, and the cell medium [65]. The images were reproduced with permission from references [57,60,61,62,63,64,65].

**Figure 10 micromachines-13-00030-f010:**
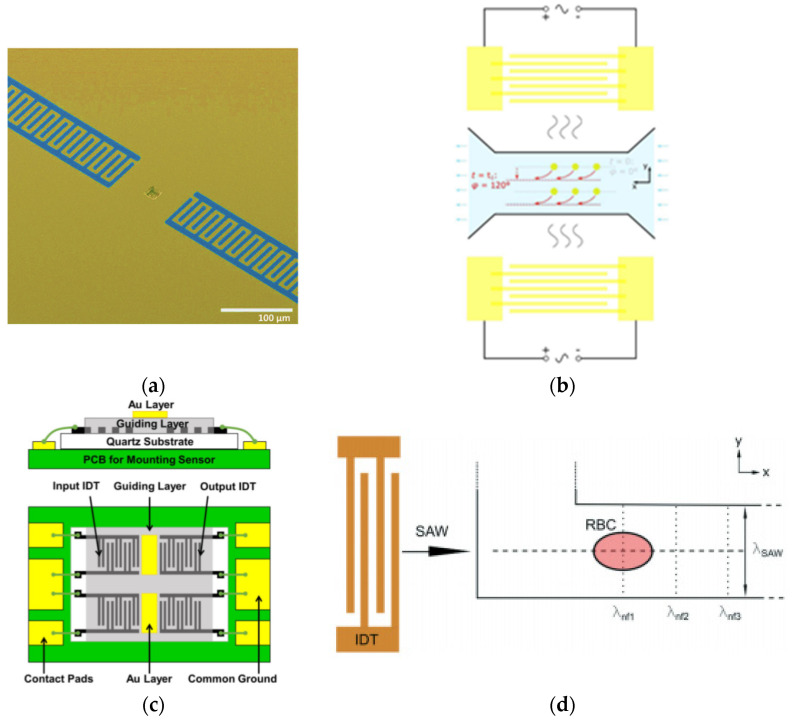
(**a**) A fabricated SAW device with a microcavity at the centre of two IDTs. (**b**) The SAW microfluidic chip and its operation. Particles are initially prefocused at the pressure node when t = 0, and when t = t1, all prefocused particles are moved to the new nodal line. (**c**) SAW device with IDTS on quartz substrate and the thin gold layer is deposited on top of the guiding layer to enhance the cell adhesion on the sensing area. (**d**) The sampling part of the acoustic erythrocytometer. Images were reproduced with permission from previous authors [78,79,80,81].

**Figure 11 micromachines-13-00030-f011:**
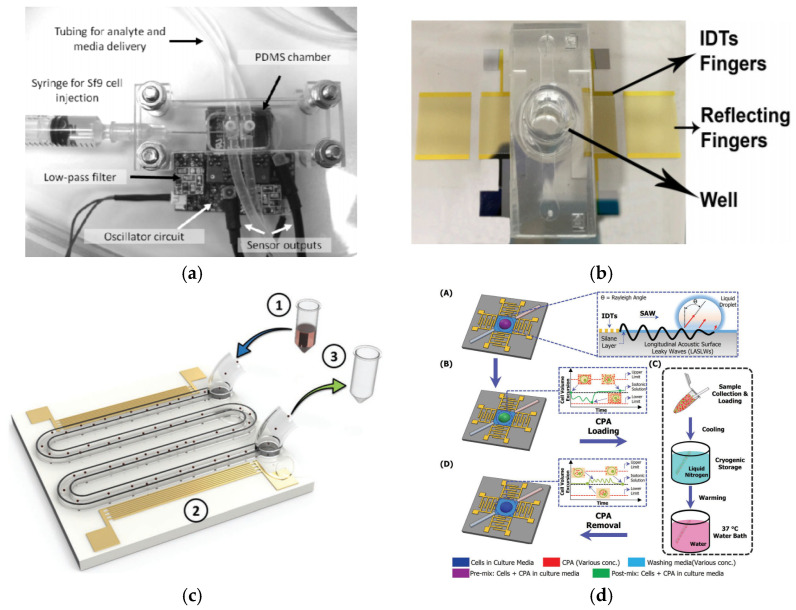
(**a**) Dual SAW biosensor with its oscillator system [84] (**b**) Fabricated SAW device with microfluidic well [86] (**c**) Design of SAW device with a serpentine channel [87] (**d**) The loading and unloading theory and details of the SAW-based CPA [88]. Images were reproduced with permission from references [84,86,87,88].

**Figure 12 micromachines-13-00030-f012:**
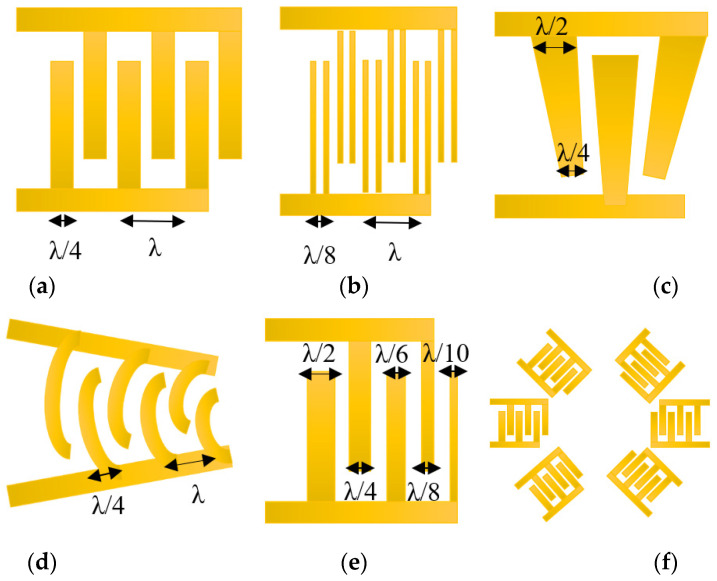
IDT used for cell analysis application (**a**) Simple IDT, (**b**) Single-Phase Unidirectional Transducer (SPUDT) IDT, (**c**) Slanted IDT, (**d**) Focused IDT, (**e**) chirped IDT, and (**f**) Multiple IDT.

**Figure 13 micromachines-13-00030-f013:**
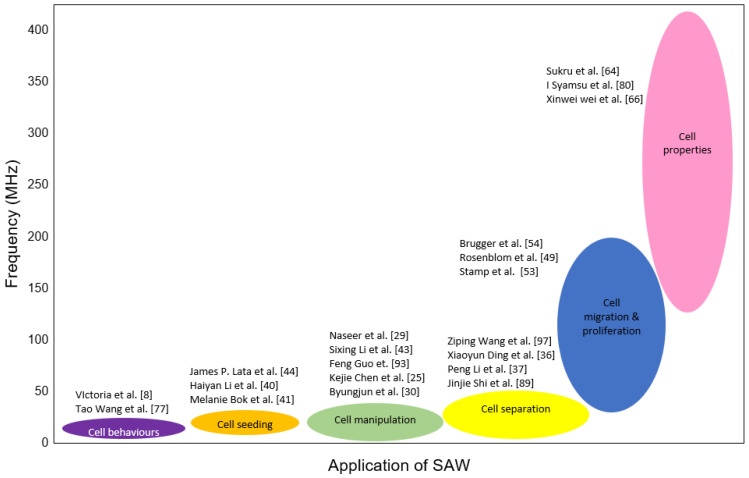
The range of acoustic wave excitation has drawn the interest of many researchers in the last decade.

**Table 1 micromachines-13-00030-t001:** Comparison of the various type interdigital (IDT) of surface acoustic wave (SAW). Images reproduced with permission from references [23,26,31,39,40,102,103].

IDT Structure/Type	Advantages	Limitation
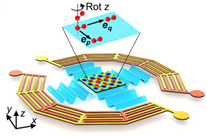 Spiral IDT [23]	For simultaneous excitation, three distinct delay paths could be used. A combination of shear horizontal SAW waves and Rayleigh SAW can be used simultaneously [103]. The 2D lattice configuration can be controlled using wavenumber–spiral acoustic tweezers.	Multi-channel function generators and high-end programmable electronics may be required.
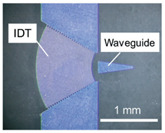 Focused IDT (FIDT) [31]	Highly focus or narrower acoustic wave beam (~50 µm wide) increases the efficiency of sorting.	To have high efficiency of sorting using FIDT, fluorescence microscopy technique should be utilized.
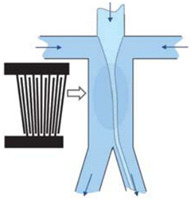 Slanted IDT [39]	Multiple frequencies can be achieved in a single IDT. Ability to cause streaming in a droplet in any direction and at any position. A sorting scheme that operates at high sorting rates of several kHz and demonstrates sorting of various cell types.	Finger width should be designed precisely, given the gap getting smaller/bigger to achieve desirable operating frequency.
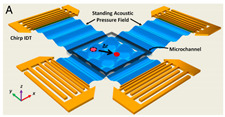 Chirped IDT [26]	Working frequency range is wider to establish a differential acoustic radiation force. Capable of driving particles to acoustic pressure field nodes or antinodes.	A limited number of electrode pairs as finger width decreases toward the target specimen.
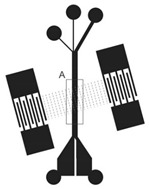 Tilted angle IDT [40]	Much wider distances of separation improve the sensitivity of separation. Utilizes separation multi pressure nodal lines. Produce a design of several nodes for increased separation efficiency.	The microfluidic channel should be positioned properly to produce a steady surface acoustic wave at an optimum angle in the direction of the fluid flow.
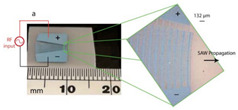 Split Phase Unidirectional Transducer (SPUDT) [102]	Generates a higher amplitude wave in one direction as compared to another direction.	Need two ports device to increase the directivity of the waves.

**Table 2 micromachines-13-00030-t002:** Summary of the various applications of IDT-based SAW device on cell studies.

Author (Year)	Working Frequency	Piezo-Electric Base	Cell Types	Applications	Outcomes	Category	Ref
Haiyan Li et al. (2007)	19.35 MHz	LiNbO_3_	Osteoblast-like cells	To increase cell seeding in scaffolds in terms of speed and consistency	The SAW-based seeding process took place in about 10 s, far faster than the seeding process (~30 min)	Cell seeding	[45]
Jinjie Shi et al. (2009)	200 MHz	LiNbO_3_	Bovine RBC and *E-coli* cells	To shape cells and microparticles	Capable of patterning any shape, size, charge, or polarity of cell or microparticle with less power than optical tweezers	Cell manipulation	[24]
Melanie Bok et al. (2009)	19.6 MHz	LiNbO_3_	Yeast cells	To seed cells without risk of denaturation	Yeast cells retain their size, morphology, and proliferation after exposure to SAW radiation, but osteoblast cells show little difference in viability.	Cell seeding	[46]
Haiyan Li et al. (2009)	10 & 20 MHz	LiNbO_3_	Osteoblast-like cells	To maintain cell viability and improve cell proliferation	Cells were delivered into the scaffold in seconds. Over 80% of the osteoblast-like cells survived after being treated with SAW at 20 MHz for 10–30 s at 380 mW.	Cell seeding	[47]
Thomas Franke et al. (2010)	140 MHz and 150 MHz	LiNbO_3_	HaCaT cells, murine fibroblasts L929 cells, MV3 melanoma cells	To sort cells continuously at high sorting rates	Single HaCaT (human keratinocytes), mouse fibroblasts, and MV3 melanoma cells are sorted at several kHz.	Cell sorting	[39]
Jinjie Shi et al. (2009)	12.6 MHz	LiNbO_3_	Microbeads (0.87 um and 4.16 um)	To sort particles into different regions based on particle volume.	In one minute, 30 mW separated 13,000 particles from a dissimilar mixture.	Cell sorting	[38]
Jinjie Shi et al. (2011)	38.2 MHz	LiNbO_3_	Microbeads	To show 3D continuous particle focusing using SSAWs in a microfluidic channel	The duration of the focusing process is about 2 s, with an input power of less than 250 mW	Cell sorting	[108]
Lothar Schmid et al. (2011)	142 MHz	LiNbO_3_	RBCs	To physiologically pump a red blood cell suspension	Simulate blood flow at 60 beats/min with an amplitude voltage modulation with a 1 Hz square wave signal	Cell manipulation	[32]
Xiaoyun Ding et al. (2012)	18.5–37 MHz	LiNbO_3_	HeLa cells	For trapping and handling of individual microparticles, cells, and organisms	Massive movement of particles up to 1600 μm/s at high speed	Cell manipulation	[26]
Zheng Tengfei et al. (2014)	2.8 MHz	LiNbO_3_		To drive nanoparticle concentration	Observed dielectrophoresis force and drag force arising to manipulate nanoparticles in a microlitre droplet using a 2.8 MHz SAW device	Cell manipulation	[109]
Sixing Li et al. (2014)	12.78 MHz	LiNbO_3_	HeLa cells, HMVEC-d cells	To create an organized cell co-culture	The SAW field sequentially patterns different cell types	Cell manipulation	[48]
Xiaoyun Ding et al. (2014)	19.4 MHz	LiNbO_3_	MCF-7 cells, WBCs	To separate MCF-7 cancer cells from healthy white blood cells	The taSSAW device works best at 2 µL/min for cells (10,000–20,000 cells/min).	Cell separation	[40]
Ninnuja Sivanantha et al. (2014)	132 MHz	LiNbO_3_	RBCs	For peeling treated RBC from a substrate and separating pathological from normal populations	A power of 500 mW delivered in 30 s provided the greatest percentage difference in cell mobilization, with healthy/treated (39%) and healthy/malaria-infected (79%)	Cell properties	[110]
Feng Guo et al. (2014)	13.35 MHz (30 mV), 13.45 MHz (10 mV)	LiNbO_3_	HEK 293T cells	For controlling the spatial distribution of cultured cells	Control the intercellular distance of cells cultured in suspension, and then convert these suspended assemblies to adherent states	Cell behaviours	[27]
Peng Li et al. (2015)	19.573 MHz	LiNbO_3_	MCF-7 cells, HeLa cells, melanoma and prostate cancer cells, UACC903M-GFP cells, and LNCaP cells	To identify CTCs in breast cancer patients’ blood samples	It is possible to recover >80% of WBCs from low concentrations of cancer cells (100 cells/mL) using this method.	Cell separation	[41]
Tao Wang et al. (2015)	14.05 MHz	LiTaO_3_	Non-cancerous (RAW 264.7) and cancer cells (A549)	To measure suspension cell mass loading and 3D cell culture platform	Relative frequency response to various cell concentrations	Cell behaviour	[86]
David J. Collins et al. (2016)	386 MHz	LiNbO_3_	NA	To overcome sorting region width limitations by using a highly focused travelling SAW.	Sorting is possible with a 25 m focused beam.	Cell separation	[99]
Sukru Ufuk Senveli et al. (2016)	196.7 MHz	Quartz	JJ012, breast cancer cell lines MDA-MB-231, SKBR3, and MCF7	To extract mechanical stiffness of cells	The elastic modulus of some cell lines differed, but the values were six orders of magnitude larger than AFM results.	Cell properties	[78]
Feng Guo et al. (2016)	13 MHz	LiNbO_3_	3T3 cells, HeLa cells	To move particles and control particle motion	Place a single cell with 1 µm precision in the x-y plane and 2 µm precision in the z-direction at 2.5 m/s.	Cell manipulation	[111]
Stamp et al. (2016)	159 MHz	LiNbO_3_	Human osteosarcoma cell line Saos-2	To improve healing rate	Application of acoustic vibrations to an artificial wound increases healing rate by 17% in vitro	Cell migration & proliferation	[57]
James P. Lata et al. (2016)	12.65 MHz	LiNbO_3_	HeLa cells, MC3T3-E1 cells, and PC12 Adh	To regulate the spatial distribution of cells and particles in hydrogel photosensitive fibres for use in tissue technology as the functional material	With an input power density (1.5 W cm^−2^) and frequency (12.65 MHz), all three polymer solutions (PEGDA 700, PEGDA 3400, and GelMA) allowed SAWs to pattern HeLa cells within the channel	Cell seeding	[49]
Shahid M Naseer et al. (2017)	3.4, 4.6, and 6.4 MHz	LiNbO_3_	Cardiomyocytes and cardiac fibroblasts	To quickly organize cells in a hydrogel-based on an extracellular matrix	Create quick, contactless cell alignment (<10 s) in gelatine methacryloyl (GelMA)	Cell manipulation	[29]
Citsabehsan et al. (2017)	81 MHz	LiNbO_3_	NA	To demonstrate microfluidic particle patterning	Describe a model that predicts the distance between patterns	Cell manipulation	[112]
Jonathan Rosenblom et al. (2017)	89 kHz	NA	CK14 cells (basal epithelial cells)	To explore the effects of low-intensity ultrasound (US) on epidermal using commercial US box, Nanochambers (NanoVibronix Inc)	In skin explants under SAW, the epidermal thickness was significantly increased compared to untreated controls	Cell migration & proliferation	[60]
Mengxi Wu et al. (2017)	20 & 40 MHz	LiNbO_3_	RBCs and WBCs	For isolating exosomes in a contact-free way from the whole blood.	Isolate whole-blood exosomes at over 99.999% of blood cell removal	Cell separation	[43]
Zhichao Ma et al. (2017)	132 MHz	LiNbO_3_	Breast cancer cell line (MCF-7)	To demonstrate a fluorescence-activated sorting system for micron-sized particles and cells in a continuous flow	Achieve highly accurate sorting with high purity (>86%) of MCF-7 cells from the target outlet	Cell separation	[113]
Kejie Chen et al. (2016)	13.35 and 13.45 MHz	LiNbO_3_	Human hepatocellular carcinoma cell, HepG2, human embryonic kidney cells, HEK 293	To yield spheroids rapidly (aggregated cells) in a high-throughput manner	Manufacture more than 150 spheroids and transfer them each 30 min to Petri plates	Cell manipulation	[28]
Byungjun Kang et al. (2018)	13.928 MHz	LiNbO_3_	HUVECs, HeLa cells	To produce a three-dimensional collateral distribution of the vessels for therapeutic vascular tissue	This shows an outstanding recovery of tissue damage	Cell manipulation	[30]
Gina Greco et al. (2018)	48.8 MHz	LiNbO_3_	Human monocytic tumour cell line U-937	To improve monocyte cell proliferation	SAW results enhanced the percentage of cell proliferation (36 ± 12) as compared to standard static cultures	Cell migration & proliferation	[61]
Mengxi Wu et al. (2018)	19.9 MHz	LiNbO_3_	PC-3, LnCaP, HeLa, and MCF-7 cancer cells	To high throughput isolate rare CTCs from blood	Cancer cells with a throughput of 7.5 mL/h isolated from leukocytes	Cell separation	[42]
Yanqi Wu et al. (2019)	12.8 MHz	LiNbO_3_	The type II alveolar epithelium-derived adenocarcinoma cell	To measure the cell compressibility and differentiate cell mechanophenotype	The compressibility of A549 cells, HASM, and MCF-7 breast cancer cells were tested and evaluated through fitting trajectories from the experiment to the equation.	Cell properties	[79]
Umar Farooq et al. (2019)	19.87 MHz	LiNbO_3_	hUCM-MSCs	To quickly transport cryoprotective products through the cell membrane	Offer high load/unload efficiency, high cell viability, and high performance.	Cell manipulation	[88]
Citsabehsan Devendran et al. (2019)	48.5 MHz	LiNbO_3_	HaCaT cells, L929 cells, MSCs, human bone marrow-derived primary cells, MG63 cells	To physically manipulate cells to influence cell biological activity	Acoustic exposure can inhibit cell adhesion, reduce cell spread and improve the metabolism of cells	Cell behaviours	[87]
Zhenhua Tian et al. (2019)	10.8, 12.1, 13.9, 20.1, 23.3 MHz	LiNbO_3_	U937 cells, HeLa cells,	For dynamic and reconfigurable manipulation of particles and cells	The cells were successfully arranged in 1D parallel and 2D rectangular lattice configurations suspended in fresh medium RPMI 1640	Cell manipulation	[22]
I Syamsu et al. (2019)	423 kHz	AlN on Si	NA	To detect cells using label-free surface acoustic wave resonators	A new chirped IDTs constructed on aluminium nitride (AlN) substrate was developed for cell detection by measuring resonant frequencies	Cell properties	[100]
Victoria Levario-Diaz et al. (2020)	6.74 MHz	PZT	Human Dermal Fibroblasts (HDF) and a cervical cancer cell line (HeLa)	To observe cell viability	Reduction in cell line metabolism after 15 min of acoustic exposure, while short acoustic exposure and slight changes in temperature and voltages have harmful effects on cells	Cell behaviours	[11]
Ziping Wang et al. (2018)	5 MHz	PZT	NA	To fabricate SAW device using silk-screen printing	The fabricated IDT-SAW device can generate standing wave fields similar to those fabricated using traditional fabrication methods.	Cell separation	[114]
Yangcheng Wang et al. (2020)	28 MHz	LiNbO_3_	L929 mouse fibroblast	To fabricate patterned microstructure using SAW for enhancing cell migration	Under SAW, cell viability and migration rate can be greatly increased.	Cell migration & proliferation	[62]
Manuel S. Brugger et al. (2020)	100 MHz	LiNbO_3_ and LiTaO_3_	MDCK-II cells, SaOs-2 cells, and T-REx-293	To study the time- and power-dependent healing of artificial wounds on a piezoelectric chip for different cell lines	Increase of the wound-healing speed of up to 135 ± 85% as compared to an internal reference	Cell migration & proliferation	[64]
Manuel S. Brugger et al. (2020)	207 MHz	LiTaO_3_	Darby Canine Kidney’ (MDCK-II)	For monitoring dynamic cell spreading and attachment	Provide cell growth information for a confluent cell layer based on the variation in the phase shift signal	Cell migration & proliferation	[63]
Xinwei Wei et al. (2020)	160 MHz	Quartz	HL-1 cardio-myocytes	To investigate HL-1 cardiomyocyte contractile properties	The cardiac contractility of different compounds can be monitored by recording the changes in insertion loss and phase shift of the SAW sensor	Cell properties	[80]
Takumi Inui et al. (2021)	100 MHz	LiNbO_3_	Mouse myoblast cell line, C2C12	To remove cells from a cell culture locally	With the input voltage of 75 V into the SAW system, approximately 12 cells are removed under SAW within a Petri dish	Cell manipulation	[31]
Lothar Schmid (2014)	161 to 171 MHz	LiNbO_3_	B16F10 mouse melanoma cells	Single-phase fluid sorting of fluorescently labelled mouse melanomas	SAW-actuated microfluidic sorter (SAWACS) with 3000/s sorting rate and 15 dBm acoustic power	Cell separation	[44]

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
