# Peer review of "Current Development in Interdigital Transducer (IDT) Surface Acoustic Wave Devices for Live Cell In Vitro Studies: A Review"

_micromachines, 2021, doi:10.3390/mi13010030_

Round 1

Reviewer 1 Report

This paper reviews  the applications of SAW based on IDT for the live-cell research including cell manipulation, separation, seeding, migration and some other behaviors. It was well written and some derails in the paper should be revised.

1, Section 2.1, Para.3 “Click or tap here to enter text.cells pattern,” Is it a typo? Also in this paragraph, what is “RBC” short for?

2, In the paper, I think “W m-2”should be W m-2

    Page 5, “mm2” should be mm2.

   “df” should be “df” and “Wn” is the same case.

    Page 11, N/m2 should be N/m2

3, In Fig.4(a), there are two additional markers (a) and (b), I think they should     be deleted. The same case is also shown in Fig.8.

Author Response

Response to Reviewer 1 Comments

First, we would like to thank the reviewers for their careful reviewing of the data presented in this manuscript and their constructive comments, which have allowed us to revise and improve the manuscript significantly.

This paper reviews the applications of SAW based on IDT for the live-cell research including cell manipulation, separation, seeding, migration and some other behaviors. It was well written and some derails in the paper should be revised.

Response: Thank you to the reviewer for the careful reviewing of our manuscript and the comments allowed us to revise and improve our manuscript significantly.

Point 1: Section 2.1, Para.3 “Click or tap here to enter text.cells pattern,” Is it a typo? Also in this paragraph, what is “RBC” short for?

Response 1: We removed Click or tap here to enter text from the paragraph as shown as follows:

Jinjie Shi et al. demonstrated one of the first SAW devices employed in cell manipulation, cells pattern, and other microparticles in a PDMS-based microchannel as shown in Figure 4 (a).

We also explained the abbreviation of RBC in the Para 3 as follows:

Jinjie Shi et al. utilised polystyrene beads, red blood cells (RBCs), and E. coli cells with different diameters to test the device adaptability using the same SAW device reported previously.

Point 2: In the paper, I think “W m-2”should be W m-2. Page 5, “mm2” should be mm2. “df” should be “df” and “Wn” is the same case. Page 11, N/m2 should be N/m2

Response 2: We revised the incorrect values and units as you suggested as follows:

In Page 3:

The researchers also discovered that the needed acoustic tweezers power intensity of 2000 W m-2 was approximately 5 x105 times lower than that of optical tweezers (109 Wm-2) [25]. 

In Page 5:

where λ is the wavelength, df is the focal distance and Wn is the aperture of the focused SAW.

In Page 11:

This finding contradicts the reports from a previous study, where the fluid recirculation led to fluid shear stress in the Petri plate (<10 mN/m2) applied to the cells and subsequently damaging the cells [14]. The cell proliferation test results on U937 cells after 48 h are shown in Figure 9 (c). SAW1 and SAW2 demonstrated different fluid shear stress at 120 mN/m2 and 280 mN/m2, respectively.

Point 3: In Fig.4(a), there are two additional markers (a) and (b), I think they should be deleted. The same case is also shown in Fig.8.

Response 3: We deleted two additional markers (a) and (b) in Fig. 4(a) and Fig. 8 as in the attachment.

Reviewer 2 Report

The scope of this study is currently a significant research area. The manuscript introduces a very nice paper focused on interdigital transducers to stimulate cell tissues. I just suggest a minor revision such that the following issue can be addressed:

1 – Section “Introduction” (1st paragraph): stimulation based on the delivery of electric fields to cell tissues has also been highly emphasized. Please see the following studies:

Bárbara M. de Sousa et al. (2021). "Capacitive interdigitated system of high osteoinductive/conductive performance for personalized acting-sensing implants", npj Regenerative Medicine, 6:80.

2 – Section “Introduction”: Fig. 1 must be quite improved.

3 – Section “Perspectives”: I suggest authors to provide a brief discussion concerning the ability of IDTs for clinical practice, mainly for multifunctional implantable bone devices. Please, I ask you to also include a suggestion of self-powering systems that hold potential to electrically supply IDTs. Suggestion: to my knowledge, hybrid electromagnetic-triboelectric generators are promising technological solutions. Please see the following literature review:

Cao, T. Zhou et al. (2017). Rotating-sleeve triboelectric electromagnetic hybrid nanogenerator for high efficiency of harvesting mechanical energy. ACS Nano 11(8), 8370.

Author Response

Response to Reviewer 2 Comments

First, we would like to thank the reviewers for their careful reviewing of the data presented in this manuscript and their constructive comments, which have allowed us to revise and improve the manuscript significantly.

The scope of this study is currently a significant research area. The manuscript introduces a very nice paper focused on interdigital transducers to stimulate cell tissues.

Response: Thank you to reviewer for the careful reviewing of our manuscript and the comments allowed us to revise and improve our manuscript significantly.

Point 1: Section “Introduction” (1st paragraph): stimulation based on the delivery of electric fields to cell tissues has also been highly emphasized. Please see the following studies:

Bárbara M. de Sousa et al. (2021). "Capacitive interdigitated system of high osteoinductive/conductive performance for personalized acting-sensing implants", npj Regenerative Medicine, 6:80.

Response 1: We added the electrical field in the 1st paragraph to include the electrical fields to cell tissues has also been highly emphasized as follows:

For example, the utilisation of biochemical, electric field, and mechanical stimuli has been extensively investigated in live-cell studies [1]-[2].

We also added the study of delivery of electric fields to cell tissues in our references:

[2]       B. M. Sousa et al., “Capacitive interdigitated delivery system of high osteogenic performance for personalized acting-sensing implantable devices,” 2021, doi: 10.1038/s41536-021-00184-6.

Point 2: Section “Introduction”: Fig. 1 must be quite improved.

Response 2: We improved the Fig. 1 by rewrite the label so that it becomes clearer as in the attachment.

Point 3: Section “Perspectives”: I suggest authors to provide a brief discussion concerning the ability of IDTs for clinical practice, mainly for multifunctional implantable bone devices. Please, I ask you to also include a suggestion of self-powering systems that hold potential to electrically supply IDTs. Suggestion: to my knowledge, hybrid electromagnetic-triboelectric generators are promising technological solutions. Please see the following literature review:

Cao, T. Zhou et al. (2017). Rotating-sleeve triboelectric electromagnetic hybrid nanogenerator for high efficiency of harvesting mechanical energy. ACS Nano 11(8), 8370.

Response 3: We added a brief discussion concerning the ability of IDTs for clinical practice and we included a suggestion of self-powering systems that hold potential to electrically supply IDTs as follows:

Lastly, despite many researchers successful to regulate cell behaviours, for instance, Haiyan Li et al. reported that primary osteoblast-like cells were delivered into the PCL scaffold in seconds under acoustic waves [45], the ability of IDT-based SAW device needs to be proved for clinical practise especially in developing multifunctional implantable bone device. Furthermore, a self-powering system is needed to electrically supply the IDTs, for example, hybrid triboelectric-electromagnetic generators is a promising technology to be integrated with SAW device to avoid bulky and complex system to power up the SAW devices.

We also included the reference regarding the self-powering systems for IDTS as follows:

[54]     R. Cao et al., “Rotating-Sleeve Triboelectric-Electromagnetic Hybrid Nanogenerator for High Efficiency of Harvesting Mechanical Energy,” ACS Nano, vol. 11, no. 8, pp. 8370–8378, 2017, doi: 10.1021/acsnano.7b03683.
